IoT-based control and monitoring system for hydroponic plant growth using image processing and mobile applications

Rofiansyah Wizman 1
Zalianty Fayza Rizka 1
La Ito Firman Ahmad 1
Wijayanto Inung 1
Ryanu Harfan Hian 1
Irawati Indrarini Dyah 2 indrarini@telkomuniversity.ac.id
1 School of Electrical Engineering, Telkom University , Bandung, West Java , Indonesia
2 School of Applied Science, Telkom University , Bandung, West Java , Indonesia
Givargis Tony
Electronic publication date: 2025 Mar 28
Publication date: 2025
Volume: 11
Electronic Location ID: e2763
Received 2024 Nov 8; Accepted 2025 Feb 24
Copyright: © 2025 Rofiansyah et al.
Copyright year: 2025
Copyright holder: Rofiansyah et al.
License: This is an open access article distributed under the terms of the Creative Commons Attribution License, which permits unrestricted use, distribution, reproduction and adaptation in any medium and for any purpose provided that it is properly attributed. For attribution, the original author(s), title, publication source (PeerJ Computer Science) and either DOI or URL of the article must be cited.
License URL: https://creativecommons.org/licenses/by/4.0/

Keywords: Urban farming, Hydroponics, Controlling, Monitoring, Mobile applications, Image processing, Internet of Things, Convolutional neural network

Funding: Director of Academic Affairs of Vocational Higher Education (DAPTV) Directorate General of Vocational Education (DIKSI) DAPTV Batch 3 Research 418/SPK/D.D4/PPK.01.APTV/VIII/2024 This work was supported by the Director of Academic Affairs of Vocational Higher Education (DAPTV), Directorate General of Vocational Education (DIKSI), for providing the DAPTV Batch 3 research grant (No. 418/SPK/D.D4/PPK.01.APTV/VIII/2024). The funders had no role in study design, data collection and analysis, decision to publish, or preparation of the manuscript.

==============================
The HydroFarm project presents an innovative IoT-based control and monitoring system for hydroponic plant growth, integrating advanced image processing techniques and mobile applications to enhance urban farming practices. This system addresses critical challenges faced by urban farmers, such as limited space and the need for precise environmental management. By employing a comprehensive approach that combines various sensors (DHT22, DS18B20, pH, TDS) with an ESP32 microcontroller, HydroFarm enables real-time monitoring of essential parameters like temperature, humidity, pH, and nutrient levels. A significant novelty of this project lies in its use of a convolutional neural network (CNN) for plant health assessment through image processing. This technique allows for accurate detection of plant conditions, categorizing leaves as healthy or unhealthy based on visual data captured via a mobile application. The application, developed in Kotlin, not only facilitates user interaction but also provides automated and manual control over nutrient delivery systems based on real-time sensor data. Testing results indicate that the HydroFarm system achieves a high accuracy rate of 96% in detecting plant health conditions, with the sensors providing accurate and consistent data to maintain effective control over hydroponic parameters. The system usability scale (SUS) evaluation yielded an impressive score of 81.875, categorizing the application as excellent and user-friendly. Overall, HydroFarm represents a significant advancement in hydroponic farming technology by integrating IoT capabilities with deep learning for enhanced decision-making and operational efficiency in urban agriculture. The findings underscore the potential for scaling this model to improve food security and promote sustainable agricultural practices in densely populated areas.

Introduction

Indonesia is a country rich in natural resources, and its people utilize these resources to meet their basic needs, particularly food, which is predominantly fulfilled through agriculture (Noer, 2019). Due to the high demand for agricultural commodities, many people work as farmers with various crops (Utami, 2022). However, over time, urban populations have gradually moved away from farming due to limited green spaces caused by urban development (Harahap, 2013). Recognizing the long-term impact of this issue, urban farming initiatives have emerged (Danugroho, 2022). Urban farming involves cultivating horticultural plants in urban areas to help meet the need for nutritious food (Khasanah, 2021). This movement was first implemented in the United States during World War I to address the economic downturn, including the rising cost of vegetables, and resulted in providing 40% of the nation’s food supply (Belinda & Rahmawati, 2017). Inspired by this, the Indonesian government has promoted urban farming as a strategy to ensure food security and build national resilience (Danugroho, 2022).

Since the pandemic, the rise in healthy lifestyles and the numerous benefits offered by vegetables have led to increased vegetable production (Halidi, 2021). Data from the Central Bureau of Statistics (BPS) shows significant fluctuations in vegetable production between 2018 and 2022, with water spinach as one example. In 2018, Indonesia produced 28.95 thousand tons of water spinach, increasing to 29.55 thousand tons in 2019. This upward trend continued in 2020 and 2021, with production reaching 31.23 thousand tons and 34.11 thousand tons, respectively. However, in 2022, production declined to 32.96 thousand tons due to a reduction in agricultural land from 62,341 to 60,991 Ha (Badan Pusat Statistik, 2023). To improve the quantity and quality of production amidst technological advancements, the government’s first policy was to encourage the younger generation to engage in agriculture (Musriadin, 2020). BPS data indicates that approximately 25.9 million or 88.52% of Indonesian farmers are aged 35 and older, while 3.36 million or 11.48% are young farmers under 35 (Bayu, 2023). This initiative involved strategies such as enhancing youth capacity in rural agriculture, developing young rural entrepreneurs, and providing access to capital (Musriadin, 2020). However, this policy faces challenges due to the lack of interest among the younger generation and prevailing stigmas associated with farming, which may hinder its success (Musriadin, 2020). As an alternative, the government introduced a second policy to boost horticultural production to meet domestic demand (Humas Direktorat Jenderal Hortikultura Kementerian Pertanian, 2023). According to the Director General of Horticulture, Prihasto Setyanto, the 7.85% increase in horticultural production prompted the government to establish three main strategies for 2021–2024: developing horticultural villages, fostering horticultural MSMEs, and digitizing agriculture through information system development (Humas Direktorat Jenderal Hortikultura Kementerian Pertanian, 2023). This policy focuses on improving seed quality and environmentally friendly management, aligning with the urban farming movement (Humas Direktorat Jenderal Hortikultura Kementerian Pertanian, 2023). Of the two policies, the government may rely on the second to meet domestic food needs through urban farming.

Urban farming can be implemented through various agricultural systems, one of which is hydroponic cultivation (Purnama, 2020). Hydroponics is an agricultural technology that uses water as a growing medium, focusing on meeting the nutritional needs of plants (Candraarya, 2021; Waluyo et al., 2021). One of the key advantages of hydroponics is its ability to be practiced in limited spaces (Roidah, 2014). In urban farming, the nutrient film technique (NFT) is a commonly used method, as it allows plants to grow in shallow water that is continuously circulated, ensuring adequate water and nutrients (Asriani, Nurcayah & Herdhiansyah, 2022; Singgih, Prabawati & Abdulloh, 2019). This method promotes faster plant growth, as the roots are in direct contact with the nutrients (Sabrina, 2022). Common plants used in hydroponic systems include pakchoi and water spinach due to their fast growth rates, high demand in urban markets, and suitability for compact spaces. These leafy vegetables thrive under controlled nutrient delivery, making them ideal for the NFT system. The nutrition required for plants in hydroponic systems is specifically tailored to their growth needs (Hutabarat et al., 2023). While the basic nutrients, like nitrogen, phosphorus, and potassium, are essential for all plants, the concentration and balance of these nutrients may vary slightly based on the type of plant (Hutabarat et al., 2023; Khozin et al., 2023). Pakchoi and water spinach, being leafy vegetables, generally require higher levels of nitrogen for healthy leaf growth. The nutrient requirements for these plants can be adjusted to ensure optimal growth in the controlled environment of a hydroponic system. Nevertheless, NFT system requires regular monitoring, as waterborne fungi can quickly spread to all plants (Sabrina, 2022). Some urban farmers have experienced crop failure due to insufficient monitoring of their hydroponic systems (Pramudita & Aprilian, 2021). Given the importance of accurately monitoring, controlling, and detecting the conditions of hydroponic plants, further study is needed on how to simplify this process for urban farmers, such as by utilizing image processing technology.

Recent studies have leveraged Internet of Things (IoT) technology for monitoring and controlling hydroponic systems (Patil et al., 2020; Agustian et al., 2022; Austria et al., 2023). The study conducted by Patil et al. (2020) focuses solely on monitoring hydroponic plants using an ESP12 microcontroller and various sensors to measure temperature, humidity, and pH levels. Meanwhile, the research presented in Agustian et al. (2022) introduces a fuzzy Mamdani inference system for managing nutrient levels in a NFT-based hydroponic system, emphasizing automated pH and total dissolved solids (TDS) control. This system employs an Arduino Mega 2560 microcontroller, ESP-01 for internet connectivity, and utilizes the Blynk platform as the user interface. Furthermore, the study in Austria et al. (2023) proposes an intelligent hydroponic system that integrates artificial intelligence (AI) algorithms to optimize nutrient management and environmental conditions, utilizing an Arduino Mega microcontroller and ESP8266 for connectivity.

Although these studies provide innovative solutions (Patil et al., 2020; Agustian et al., 2022; Austria et al., 2023), certain gaps remain, particularly in terms of limited user interaction, as they do not sufficiently explore user interface interactions or mobile applications for end-users. Additionally, none of these studies investigate deep learning techniques such as convolutional neural networks (CNN) for plant health assessment. To address these limitations, HydroFarm introduces a comprehensive mobile application developed using Kotlin, enabling users to monitor environmental data and control pumps either manually or automatically. Moreover, HydroFarm incorporates CNN-based plant health detection, leveraging state-of-the-art technology to provide accurate plant condition assessments. Finally, the system’s ability to adjust nutrient levels based on real-time sensor data and user input creates a responsive environment conducive to optimal plant growth, distinguishing HydroFarm from existing solutions.

Materials and Methods

The block diagram of HydroFarm, as shown in Fig. 1, consists of several interconnected components designed to control and monitor the growth of hydroponic plants. HydroFarm is equipped with various sensors to assess the nutrient levels in the hydroponic setup. The sensors used include the DHT sensor for measuring temperature and humidity, the DS18B20 sensor for monitoring water temperature, a pH sensor for determining the acidity/alkalinity of the water, and a TDS sensor to measure the total dissolved solids or particles in the water. The data from these sensors are processed by the ESP32 microcontroller, which displays the information on an LCD and sends it to Firebase. The collected data in Firebase from the ESP32 can be monitored by the user. Additionally, users can control the pumps through an integrated application that connects Firebase to the ESP32, enabling them to have full control over the conditions of the hydroponic plants anytime and anywhere. The application is equipped with features to detect the conditions of the hydroponic plants using a model trained with a CNN in the form of TensorFlow Lite (tflite). The mobile application is developed using the Kotlin programming language. Users can also modify the input values for TDS and pH through the application, which are then sent to Firebase to reflect the changes, allowing the ESP32 to adjust the TDS and pH levels accordingly.

Figure 1 HydroFarm implementation block diagram.

Internet of Things system for controlling and monitoring the growth of hydroponic

The implementation of the Internet of Things (IoT) in a smart hydroponic system is designed to facilitate the monitoring and control of various devices or systems automatically and efficiently to support the growth of hydroponic plants. To create an accurate and precise system, a series of technical processes are required, from calibration to component integration. The following are the main components involved in the IoT implementation in HydroFarm, as illustrated in Fig. 2.

Figure 2 Wiring of IoT devices for HydroFarm system.

The IoT HydroFarm system consists of the following components: (A) ESP32 as the microcontroller, responsible for processing and transmitting data; (B) a power supply that ensures stable energy distribution; (C) a DHT22 sensor, which measures air temperature and humidity within the environment; (D) a DS18B20 sensor, used to monitor water temperature; (E) a pH sensor that determines the acidity or alkalinity level of the nutrient solution; (F) a TDS sensor, which quantifies the concentration of dissolved solids in water, ensuring optimal nutrient balance; (G) a relay module and water pump, which regulate the flow of nutrients by controlling water circulation; and (H) an LCD I2C display, which provides real-time local monitoring of sensor data for improved system management.

The wiring process of components, as shown in Fig. 2, is necessary to assist the builder in determining the number of wires and creating the wiring paths. The printed circuit board (PCB) design in Fig. 3 is intended to streamline the shape of the device, with the design adjusted according to the component wiring. This design enhances system compactness, minimizes signal interference, and improves electrical stability, thereby increasing overall reliability and performance. Ultimately, it streamlines hardware assembly while contributing to the efficiency and effectiveness of the IoT-based hydroponic system.

Figure 3 PCB design for Internet of Things.

The PCB design optimizes component layout to enhance compactness, reduce signal interference, and improve electrical stability.

To build an IoT-based smart hydroponic monitoring and controlling system, several essential components are required:

ESP32-DevKitC V4

The ESP32-DevKitC V4 is a small microcontroller/development board based on the ESP32, manufactured by ESPRESSIF (ESPRESSIF, 2025). The ESP32 microcontroller acts as the central controller, processing sensor data and managing communication between sensors, actuators, and the cloud.

In HydroFarm, the ESP32 serves as the brain of the subsystem, receiving data from the DHT sensor, DS18B20 sensor, pH sensor, and TDS sensor, which are then processed and sent to Firebase. The pump is controlled by the ESP32 based on data from the DS18B20, pH, and TDS sensors, with predetermined upper and lower threshold values, ensuring that the hydroponic plants consistently receive the appropriate amount of nutrients.

DHT22 sensor

The DHT22 is a sensor used to measure temperature and humidity in the surrounding environment of the hydroponic system. This sensor provides a calibrated digital output signal, ensuring highly accurate readings, reliable sensing, and excellent long-term stability (Saputra, Suchendra & Sani, 2020).

The DHT22 sensor, placed inside the IoT box, features a small volume, very low power consumption, a signal transmission range of up to 20 m, and the ability to measure temperatures from −40~80 °C with an accuracy of ±0.5 °C, and humidity levels from 0 to 100% with an accuracy of ±5%. These characteristics make it an ideal choice for monitoring temperature and humidity in a hydroponic plant system (Saputra, Suchendra & Sani, 2020).

DS18B20 sensor

The DS18B20 sensor monitors water temperature to ensure optimal conditions for plant growth, making it an ideal choice for measuring water temperature in hydroponic systems when placed in the nutrient reservoir (DFROBOT, 2017). It is designed for wet conditions, providing accurate digital output and long-term stability. The sensor uses a capacitive humidity sensing element and an NTC temperature measurement element, connected via a microcomputer chip. It can be configured to 9–12 bit resolution through a single-wire digital pin and measures temperatures ranging from −55 °C to 125 °C (−67 °F to 257 °F) with an accuracy of ±0.5 °C from −10 °C to 85 °C (DFROBOT, 2017).

PH SEN0161-V2 sensor

The pH sensor (potential of hydrogen) SEN0161-V2 is used to measure the pH level or the acidity/alkalinity of a liquid or solution. The core functionality of the pH water sensor lies in its probe, which is made of a glass electrode (Pramesia Pratama, Wibawa & Suarjaya, 2022). The glass electrode contains an HCl solution at the tip, which measures the concentration of H3O+ ions in a solution, determining the pH level (Pramesia Pratama, Wibawa & Suarjaya, 2022). The electrode is highly sensitive, with low impedance, enabling fast and stable readings in both high and low-temperature liquids (Pramesia Pratama, Wibawa & Suarjaya, 2022). The sensor provides analog output readings that can be interfaced with a microcontroller through a built-in PH 2.0 interface on the pH module (Pramesia Pratama, Wibawa & Suarjaya, 2022). This sensor is highly effective for long-term pH monitoring in liquids, making it an ideal choice for measuring the pH of hydroponic water.

TDS SEN0244 sensor

The total dissolved solids (TDS) sensor SEN0244 measures the concentration of dissolved solids in water, such as salts, minerals, metals, and other compounds. The TDS sensor works by measuring the electrical conductivity of the water using two probes immersed in the solution. The sensor then processes the conductivity and outputs an analog signal representing the TDS level (Irawan et al., 2021). Placed in the nutrient reservoir, this sensor measures the concentration of nutrient solutions in hydroponic systems in parts per million (ppm), allowing precise control of the hydroponic nutrient levels to ensure optimal plant growth.

Relay module and water pump

A relay is an electrically activated switch and an electromechanical component consisting of two main elements: an electromagnet (coil) and a mechanical switch. The operating principle of a relay is based on the use of an electromagnet to control the switch. This switch is operated by a low voltage to direct the flow of current at a higher voltage. The primary function of the relay is to connect and disconnect electrical circuits, enabling effective and safe control of electrical devices with different voltage and current levels. In HydroFarm, the SRD-05VDC-SL-C relay serves as a smart switch to control pumps that regulate nutrient flow for hydroponic plants. This relay operates with a 5V DC input voltage and is triggered by a low control voltage. It has a load capacity of up to 10A at 250 VAC or 125 VAC for AC currents and 10A at 30 VDC or 28 VDC for DC currents, enabling efficient control of high-power devices.

LCD I2C 20 × 4

To display sensor data, an LCD I2C 20 × 4 is required. The liquid crystal display (LCD) I2C is a type of display that uses the I2C (Inter-Integrated Circuit) communication protocol to interact with a microcontroller or other devices. I2C is a communication protocol that allows electronic devices to communicate with each other via shared data and clock signal lines. This protocol enables multiple devices to connect on the same bus, with each device having a unique address. The LCD I2C 20 × 4 provides a clear way to locally monitor real-time data from sensors.

Convolutional neural network for detecting plant health or conditions

In HydroFarm, image processing is employed to detect the condition of hydroponic plants, specifically focusing on the leaves. This process involves developing a model through training data and implementing a detection system in a mobile application. A common neural network architecture used for image analysis is the CNN, which simplifies images to provide more accurate analysis results (Desai & Shah, 2021). CNN consists of key layers such as convolutional layers, pooling layers, and fully connected layers, influenced by parameters like input, filter, padding, stride, pooling, weight, and bias (Mulyawan, 2025).

Using the VGG16 for leaf health detection on a small dataset can be justified due to its robust feature extraction capabilities. VGG16 comprises 13 convolutional layers, three fully connected layers, and five pooling layers, and is pre-trained on the ImageNet dataset. Its weights are fine-tuned and extended with custom layers to classify leaf conditions into two categories: healthy and unhealthy. By leveraging pretrained weights, VGG16 adapts well to small datasets without requiring extensive data or training time, reducing the risk of overfitting. Optimized for mobile applications, the model is converted to TensorFlow Lite format for efficient deployment (Yang et al., 2021). Its well-established performance and ease of implementation make it a reliable baseline model for benchmarking and further optimization, despite the added complexity.

The dataset used in this project includes images of pakchoi and water spinach, consisting of 371 images categorized into two classes, resized to 240 × 240 pixels to match CNN model’s input requirements and split into training, testing, and validation sets. To enhance generalization and reduce overfitting, data augmentation is applied using Image Data Generator with transformations including rescaling pixel values to range [0, 1], shear distortions, horizontal and vertical flips, brightness adjustment (50% to 150%), random rotation (up to 40 degrees), and zooming (20%). The training process is configured with batch size of 15, 50 epochs, and dynamically computed staps per epoch and validation steps based on dataset size. ADAM optimizer with learning rate 0.001 is used alongside categorical cross-entropy loss function, with ReLU activation for intermediate layers and Softmax for output layer to handle multi-class classification. After training, the model’s performance is evaluated based on validation accuracy with the best epoch recorded for further analysis.

In the implementation process, as shown in Fig. 4, the process begins with collecting a dataset of hydroponic plant leaf images categorized by their condition (healthy and unhealthy) over a 15-day period, which is divided into three sets with distribution ratios of 70% for training, 15% for testing, and 15% for validation in the first test; and 80% for training, 10% for testing, and 10% for validation in the second test.

Figure 4 Block diagram of CNN training process.

Next, preprocessing is performed to enhance image quality, including rescaling (1/255), shear transformation (0.2), horizontal and vertical flips, brightness adjustment ([0.5, 1.5]), rotation (40), and zooming (0.2). After preprocessing, the training dataset is used to train the CNN model by adjusting parameters such as learning rate and batch size, followed by evaluating the model’s performance using the testing and validation datasets to ensure accuracy exceeds 85%. If the performance criteria are not met, analysis, debugging, and hyperparameter tuning are conducted before retraining the model. Once the criteria are met, the model is saved in .tflite format and is ready to be implemented for detecting hydroponic plant conditions.

Figure 5 illustrates the process of implementing the CNN model in HydroFarm, beginning with capturing or uploading an image of a hydroponic leaf. The image undergoes preprocessing, where it is resized to 240 × 240 pixels to match the required input shape for the CNN model. The preprocessed image is then fed into CNN model for analysis. If the model cannot detect a leaf in the input image, an error message is displayed, prompting the user to upload another image. If a leaf is successfully detected, the model proceeds with classifying the leaf’s condition. The classification results are then displayed to the user through the system interface. The CNN model is implemented using a Sequential architecture and optimized with the Adam optimizer. The dataset for training and evaluation includes primary data collected from hydroponic farmers camera, as well as secondary data sourced from Google, focusing on the analysis of green pigments to detect the condition of hydroponic plants via mobile application.

Figure 5 Block diagram of CNN implementation on mobile apps.

Mobile application as a user interface for monitoring and controlling IoT and CNN integration

The development of the HydroFarm application interface, as shown in Fig. 6, was conducted using Kotlin, the native programming language for Android application development. For this development process, Android Studio was utilized as the primary integrated development environment (IDE). Android Studio was chosen for its comprehensive support for Android application development, including features for debugging, user interface (UI) visualization, and device simulation through the Android Emulator.

Figure 6 UI/UX design of HydroFarm application.

The HydroFarm application is designed to integrate two main components: the deep learning (DL) model implemented using TensorFlow Lite (.tflite) and real-time sensor data received from Firebase Realtime Database. The DL model is used to detect the condition of plants based on the provided input, while Firebase serves as the central storage and management hub for sensor data from the IoT devices installed in the hydroponic system.

Results and Discussion

Testing is a process aimed at determining the performance of HydroFarm, as shown in Fig. 7, to ensure it operates optimally, accurately, and consistently according to the established specifications and objectives. The system testing is conducted directly on the Smart Hydroponic devices. During a 1-week testing period in the laboratory, various critical aspects of HydroFarm will be evaluated to ensure its performance aligns with the established specifications.

Figure 7 HydroFarm hardware devices.

First, testing will validate the system’s ability to provide information on temperature and humidity. Various scenarios will be run to measure the precision and accuracy of the sensors installed in the system, ensuring that any changes in environmental conditions can be detected and reported in real time. Second, the TDS aspect will be tested by measuring the total concentration of dissolved substances in the nutrient solution. The collected data will be analyzed to determine whether the system can provide TDS information with the required accuracy.

Third, HydroFarm will be tested for its capability to provide information on the water pH. Various nutrient solutions with different pH levels will be prepared and tested, with results recorded and compared to evaluate the system’s accuracy. Fourth, the system will also be tested regarding its ability to provide water temperature information. Water temperature is a vital parameter for determining optimal conditions for hydroponic plant growth, making its accuracy essential for ensuring plant development (Pasandaran, Djufry & Suradisastra, 2019).

In addition to environmental parameters, HydroFarm will be tested for its ability to detect the condition of hydroponic plant leaves using CNN. Furthermore, data from all integrated sensors will be successfully displayed in real time on the Smart Hydroponic application. Testing will be conducted to ensure the integrity and smooth flow of data between the system and the application, providing users with accurate information.

In terms of control, HydroFarm will be evaluated for its ability to manually and automatically control the pumps through the Smart Hydroponic application. The entire automation process will be tested under various scenarios to ensure the system’s reliability and responsiveness. Finally, with a focus on display and user interface, the system will be tested to ensure that information regarding hydroponic plant diseases is presented clearly and understandably in the Smart Hydroponic application, facilitating users in making informed decisions or taking necessary further actions.

Results of the Internet of Things system

The following is an overview of the testing of the IoT components for the control and monitoring system of hydroponic plant growth.

ESP32-DevKitC V4

The ESP32 operates effectively as the central controller, processing sensor data and managing communication between sensors, actuators, and the cloud. The ESP32 is able to drive all the sensors and pumps simultaneously, as demonstrated by the successful testing of the other sensors/modules and the effective transmission of data to the Firebase realtime database, as shown in Fig. 8.

Figure 8 Firebase realtime database framework for HydroFarm.

DHT22 sensor

The testing of air temperature and humidity enviroment involved integrating a DHT22 sensor with a microcontroller, supplying power to the system, and programming the microcontroller to activate the sensor’s functionality. The device was positioned near hydroponic plant to observe and analyze the temperature and humidity data provided by the sensor. The performance results of the DHT22 sensor in measuring air temperature and humidity enviroment, as detailed in Table 1, demonstrate its accuracy in providing reliable environmental data, effectively reflecting actual conditions, including its response when exposed to fire.

Table 1 DHT22 sensor testing results for measuring temperature and humidity.

No.	Room condition	
Normal	Near the fire	
Temperature	Humidity	Temperature	Humidity	
1.	27.0 °C	77%	27.1 °C	79%	
2.	27.0 °C	77%	28.5 °C	74%	
3.	27.0 °C	77%	29.1 °C	80%	
4.	27.0 °C	77%	29.8 °C	83%	
5.	27.0 °C	77%	31.8 °C	84%	

DS18B20 sensor

The testing of water temperature involved integrating the DS18B20 sensor with a microcontroller, supplying power to the system, and programming the microcontroller to activate the sensor’s functionality. The device was positioned near hydroponic plant to observe and analyze the water temperature data provided by the sensor. The performance results of the DS18B20 sensor in measuring water temperature, as detailed in Table 2. These findings demonstrate that the DS18B20 sensor operates effectively, accurately reflecting the water temperature under actual conditions, including its response when the water with or without ice.

Table 2 DS18B20 sensor testing results for measuring air temperature.

No.	Water temperature	
Normal	With ice	
1.	25.8 °C	12.3 °C	25.8 °C	12.3 °C	
2.	25.8 °C	10.1 °C	25.8 °C	10.1 °C	
3.	25.8 °C	8.11 °C	25.8 °C	8.11 °C	
4.	25.8 °C	6.71 °C	25.8 °C	6.71 °C	
5.	25.8 °C	5.71 °C	25.8 °C	5.71 °C	

pH SEN0161-V2 sensor

The evaluation of water pH measurement involved integrating the pH sensor with a microcontroller, supplying power to the system, and programming the microcontroller to activate the sensor’s functionality. The device was positioned near the hydroponic plant to observe and analyze the water pH data provided by the sensor, assessing its accuracy and determining the need for calibration. If calibration was required, it was performed by reprogramming the system using a linear regression formula, followed by data analysis of the observations.

After conducting tests, the following are the results from the pH sensor in providing water pH information when the probe was immersed in calibration solutions of pH 4.00 (25 °C), pH 6.86 (25 °C), and pH 9.18 (25 °C), as shown in Tables 3–5.

Table 3 pH sensor testing results using pH 4.00 solution before calibration.

No.	Calibration solution pH 4.00 (25 °C)
Before calibration	
pH sensor	pH tester	Difference in pH value	Percentage
error pH	
1.	2.88	4.00	1.12	38.89%	
2.	2.88	4.00	1.12	38.89%	
3.	2.88	4.00	1.12	38.89%	
4.	2.88	4.00	1.12	38.89%	
5.	2.88	4.00	1.12	38.89%	
Average	2.88	4.00	1.12	38.89%	

Table 4 pH sensor testing results using pH 6.86 solution before calibration.

No.	Calibration solution pH 6.86 (25 °C)
Before calibration	
pH sensor	pH tester	Difference in pH value	Percentage
error pH	
1.	2.23	6.86	4.63	207.62%	
2.	2.23	6.86	4.63	207.62%	
3.	2.23	6.86	4.63	207.62%	
4.	2.23	6.86	4.63	207.62%	
5.	2.23	6.86	4.63	207.62%	
Average	2.23	6.86	4.63	207.62%	

Table 5 pH sensor testing results using pH 9.18 solution before calibration.

No.	Calibration solution pH 9.18 (25 °C)
Before calibration	
pH sensor	pH tester	Difference in pH value	Percentage
error pH	
1.	1.68	9.18	7.5	446.43%	
2.	1.68	9.18	7.5	446.43%	
3.	1.68	9.18	7.5	446.43%	
4.	1.68	9.18	7.5	446.43%	
5.	1.68	9.18	7.5	446.43%	
Average	1.68	9.18	7.5	446.43%	

From the results of the above tests, there is a significant discrepancy between the values measured by the pH sensor and the calibration solutions used as a comparison. Since the test results do not match the expected values of the solutions, calibration of the pH sensor coding is required. The calibration method used is a linear regression approach. This linear regression approach is necessary to minimize the difference between the pH sensor readings and the solution values. The formula obtained is shown in Fig. 9.

Figure 9 Linear regression graph of pH sensor calibration values.

The testing was conducted again with the pH sensor that had been calibrated using the linear regression formula. Below are the results from the calibrated pH sensor testing while the probe was immersed in calibration solutions at pH 4.00 (25 °C), pH 6.86 (25 °C), and pH 9.18 (25 °C), as shown in Tables 6–8. From these tables, it is shown that the data remained consistent, indicating that the pH sensor is capable of providing consistent and sufficiently accurate information, as evidenced by its small percentage error.

Table 6 pH sensor testing results using pH 4.00 solution after calibration.

No.	Calibration solution pH 4.00 (25 °C)
After calibration	
pH sensor	pH tester	Difference in pH value	Percentage
error pH	
1.	4.10	4.00	0.1	2.44%	
2.	4.10	4.00	0.1	2.44%	
3.	4.10	4.00	0.1	2.44%	
4.	4.10	4.00	0.1	2.44%	
5.	4.10	4.00	0.1	2.44%	
Average	4.10	4.00	0.1	2.44%	

Table 7 pH sensor testing results using pH 6.86 solution after calibration.

No.	Calibration solution pH 6.86 (25 °C)
After calibration	
pH sensor	pH tester	Difference in pH value	Percentage
error pH	
1.	6.90	6.86	0.04	0.58%	
2.	6.90	6.86	0.04	0.58%	
3.	6.90	6.86	0.04	0.58%	
4.	6.90	6.86	0.04	0.58%	
5.	6.90	6.86	0.04	0.58%	
Average	6.90	6.86	0.04	0.58%	

Table 8 pH sensor testing results using pH 9.18 solution after calibration.

No.	Calibration solution pH 9.18 (25 °C)
Before calibration	
pH sensor	pH tester	Difference in pH value	Percentage
error pH	
1.	9.20	9.18	0.02	0.217%	
2.	9.20	9.18	0.02	0.217%	
3.	9.20	9.18	0.02	0.217%	
4.	9.20	9.18	0.02	0.217%	
5.	9.20	9.18	0.02	0.217%	
Average	9.20	9.18	0.02	0.217%	

TDS SEN0244 sensor

The evaluation of TDS measurement involved integrating the TDS sensor with a microcontroller, supplying power to the system, and programming the microcontroller to activate the sensor’s functionality. The device was positioned near the hydroponic plant to observe and analyze the TDS data provided by the sensor, assessing its accuracy and determining the need for calibration. If calibration was required, it was performed by reprogramming the system using a linear regression formula, followed by data analysis of the observations.

After conducting the tests, the following are the results from the TDS sensor in providing total dissolved solids information when the probe was immersed in calibration solutions of 342 ppm (25 °C) and 500 ppm (25 °C), as shown in Tables 9 and 10.

Table 9 TDS sensor testing results using 342 ppm calibration solution (25 °C) before calibration.

No.	Calibration solution 342 ppm (25 °C)
Before calibration	
TDS sensor	TDS tester	Difference in TDS value	Percentage
error TDS	
1.	138	342	204	147.826%	
2.	138	342	204	147.826%	
3.	138	342	204	147.826%	
4.	138	342	204	147.826%	
5.	138	342	204	147.826%	
Average	138	342	204	147.826%	

Table 10 TDS sensor testing results using 500 ppm calibration solution (25 °C) before calibration.

No.	Calibration solution 500 ppm (25 °C)
Before calibration	
TDS sensor	TDS yester	Difference in TDS value	Percentage
error TDS	
1.	195	500	305	156.41%	
2.	195	500	305	156.41%	
3.	195	500	305	156.41%	
4.	195	500	305	156.41%	
5.	195	500	305	156.41%	
Average	195	500	305	156.41%	

From the test results above, there is a significant difference between the values measured by the TDS sensor and the calibration solution values used for comparison. Since the test results did not align with the solution values, calibration of the TDS sensor coding is necessary. The calibration approach used is linear regression. This linear regression approach is essential to minimize the discrepancy between the TDS sensor values and the solution values. The formula obtained is shown in Fig. 10.

Figure 10 Linear regression graph of TDS sensor calibration values.

The testing was conducted again using the TDS sensor that had been calibrated with the linear regression formula. The results of the calibrated TDS sensor testing, while the probe was immersed in calibration solutions of 342 ppm (25 °C) and 500 ppm (25 °C), are shown in Tables 11 and 12. From these tables, it is shown that the data remained consistent, indicating that the TDS sensor is capable of providing consistent and sufficiently accurate information, as evidenced by its small percentage error.

Table 11 TDS sensor testing results using 342 ppm calibration solution (25 °C) after calibration.

No.	Calibration solution 342 ppm (25 °C)
After calibration	
TDS sensor	TDS tester	Difference in TDS value	Percentage
error TDS	
1.	345	342	3	0.87%	
2.	345	342	3	0.87%	
3.	345	342	3	0.87%	
4.	345	342	3	0.87%	
5.	345	342	3	0.87%	
Average	345	342	3	0.87%	

Table 12 TDS sensor testing results using 500 ppm calibration solution (25 °C) after calibration.

No.	Calibration solution 500 ppm (25 °C)
After calibration	
TDS sensor	TDS tester	Difference in TDS value	Percentage
error TDS	
1.	508	500	8	1.575%	
2.	508	500	8	1.575%	
3.	508	500	8	1.575%	
4.	508	500	8	1.575%	
5.	508	500	8	1.575%	
Average	508	500	8	1.575%	

Relay module and water pump

After a series of tests, the Relay module and water pump demonstrated satisfactory performance. The system was successfully configured to function automatically, with the Relay module able to activate or deactivate the water pump based on pH and TDS measurements received from the sensors. With the parameters set in the application, the water pump can operate automatically flowing water when the pH and TDS values are below or above the predetermined limits, and conversely, shutting off the water flow when conditions are within the ideal range. Additionally, the system is equipped with a manual feature, allowing users to control the water pump directly through the application. By simply pressing the available on/off button, users can activate or deactivate the water pump as needed, providing extra flexibility in managing the hydroponic plants.

In addition, the system is equipped with nutrient pumps, specifically Pump A Nutrient (for macro-nutrients like N (nitrogen), P (phosphorus), K (potassium), and Mg (magnesium)) and Pump B Nutrient (for micro-nutrients like Fe (iron), Mn (manganese), B (boron), Cu (copper), Zn (zinc), Cl (chlorine), Si (silicon), Na (sodium), and Co (cobalt)), which are activated based on TDS sensor readings. These pumps deliver the appropriate nutrients according to the needs of the plants, ensuring optimal nutrient levels. Furthermore, the system includes pH regulation features, with the pH up pump and pH down pump adjusting the pH levels as indicated by the pH sensor. This ensures that the hydroponic system maintains an ideal pH range for plant growth.

LCD I2C 20 × 4

After implementation and testing, the LCD demonstrated good performance, as shown in Fig. 11, accurately and clearly displaying data on TDS, pH, water temperature, air temperature, and humidity. Users can easily monitor environmental conditions, allowing for the evaluation of water quality and timely actions to maintain ideal parameters for plant growth.

Figure 11 LCD display of HydroFarm hardware devices.

Information regarding water and air temperatures helps users identify changes that could affect plant growth, enabling quick adjustments to the temperature control system. Additionally, the real-time displayed humidity data provides crucial insights to prevent stress on the plants and maintain their overall health. Overall, this system supports more effective monitoring and management of the environment within the Smart Hydroponic system.

Results of convolutional neural network

Testing was conducted using two scenarios for splitting the data into training, testing, and validation sets: 70%, 15%, and 15% for the first test, and 80%, 10%, and 10% for the second test. In the first test, the resulting model achieved an accuracy of 96.1%, with precision, recall, and F1-score values shown in Fig. 12, along with a computation time of 10 s.

Figure 12 Precision, recall, F1-score, and accuracy values from first test.

Next, a check was performed on the test dataset, and the classification results were mapped in the form of a confusion matrix. Figure 13 shows the confusion matrix for classifying hydroponic plant leaves, where out of 38 healthy leaf datasets, 36 were classified as healthy and two were classified as unhealthy. For the 39 unhealthy leaves, 38 were classified as unhealthy, while one dataset was incorrectly classified as healthy.

Figure 13 Confusion matrix from first test.

In addition to testing the data against the model, validation testing was conducted to ensure that the created model does not suffer from overfitting or underfitting by comparing the test accuracy results with the validation accuracy. Figure 14 shows that the first test has a test accuracy of 96.1% and a validation accuracy of 98.78%. The validation dataset accuracy is higher than the test dataset accuracy; however, the small difference indicates that the model performs well on the test dataset and does not exhibit significant overfitting.

Figure 14 Validation accuracy and test accuracy values from first test.

In the second test, the resulting model achieved an accuracy of 90.38%, along with precision, recall, and F1-score values as shown in Fig. 15, with a computation time of 6 s.

Figure 15 Precision, recall, F1-score, and accuracy values from second test.

Next, a check was performed on the test dataset, and the classification results were mapped in the form of a confusion matrix. Figure 16 shows the confusion matrix for the classification of hydroponic plant leaves, where out of 26 healthy leaf datasets, 23 were classified as healthy and three as unhealthy. Meanwhile, for the 26 unhealthy leaves, 24 were classified as unhealthy and two as healthy.

Figure 16 Confusion matrix from second test.

In addition to testing the model against the test dataset, validation data testing was conducted to ensure that the created model does not experience overfitting or underfitting by comparing the test accuracy results with the validation accuracy. Figure 17 shows that the first test had a test accuracy of 90.38% and a validation accuracy of 100%. Among these values, the accuracy on the validation dataset is higher than that on the test dataset. The validation dataset accuracy is very high, reaching 100%, indicating that the model fits the validation data exceptionally well. However, an accuracy of 100% on the validation dataset may indicate potential overfitting on a relatively small dataset.

Figure 17 Validation accuracy and test accuracy values from second test.

By comparing both scenarios, the model from the first test is deemed better for subsequent testing. After conducting tests with the previously designed scenarios, the model can detect the condition of hydroponic plant leaves, accompanied by confidence values. For hydroponic plants with green leaf characteristics, the model detects them as healthy plants with an accuracy of 93.56%. For hydroponic plants with yellowing leaves, the model classifies them as unhealthy plants with an accuracy of 51.84%. Additionally, testing on wilted pakchoi plants resulted in the model detecting them as unhealthy plants with an accuracy of 99.62%. The testing was conducted directly using the application, as shown in Fig. 18.

Figure 18 Testing for hydroponic plant condition detection.

Result of mobile application

In an effort to develop an IoT-based smart hydroponic system integrated with a Convolutional Neural Network (CNN) model for image processing, an effective and intuitive application is needed. This application is designed with three main menus: Home, Take Photo, and Profile.

The Home menu in the developed smart hydroponic application serves as the main interface, where users can monitor environmental data in real-time, including TDS, pH, water temperature, air temperature, and humidity enviroment. This visual information is designed to facilitate users in making decisions regarding the management of the hydroponic system. This menu is equipped with control features that allow users to manage the nutrient pump both manually and automatically, providing enhanced flexibility in adjusting nutrient levels based on the specific requirements of different plant varieties.

In the automatic control feature, as shown in Fig. 19, users can set the desired pH and TDS range; after entering these values, the system will automatically adjust the operation of the nutrient pump according to the established parameters. For example, if the pH or TDS values fall outside the range inputted by the user, the nutrient pump will be activated or deactivated automatically to maintain stable conditions for the plants, allowing the system to function autonomously and responsively to changes in the environment. Additionally, the application also provides a manual control option, as shown in Fig. 20, where users can press the provided button to turn the nutrient pump on or off directly according to their preferences. This feature offers added flexibility for users who want to take control and make adjustments based on their specific needs at that moment.

Figure 19 Home screen display of HydroFarm automated menu.

Figure 20 Home screen display of HydroFarm manual menu.

The Take Photo menu, as shown in Fig. 21, integrated within the HydroFarm application, provides users with the capability to capture plant images directly using their smartphone camera or select images from the gallery. This feature is designed to facilitate the collection of visual data, which is then processed by the CNN model embedded in the application. The use of this technology is crucial for detecting plant health conditions more accurately and efficiently.

Figure 21 Take photo screen display of HydroFarm.

Once the plant image is captured or uploaded, the CNN model analyzes the image to detect various health indicators of the plant. This CNN-based analysis enables early identification of potential issues, allowing users to take corrective actions promptly before further damage occurs.

The Profile menu, as shown in Fig. 22, in the developed smart hydroponic application, is designed to provide educational information about the HydroFarm app and guidelines for its usage. This feature aims to help users gain a deeper understanding of the app’s core concept and primary objectives, as well as how to operate it effectively for managing their hydroponic system.

Figure 22 Profile screen display of HydroFarm.

To determine the feasibility of the HydroFarm application, the researcher used the system usability scale (SUS) method. SUS, created by John Brooke (Huda et al., 2023), is a measurement tool used to assess the functionality of a product by testing it directly with users. This method is beneficial for evaluating an application based on its features and usability from the user’s perspective.

The SUS method was carried out via a Google Form containing 10 questions rated on a five-point scale, ranging from “Strongly Disagree” to “Strongly Agree.” Specifically, there are five positive and five negative questions. The 10 questions are as follows:

1. I feel comfortable using HydroFarm.

2. I find the features in HydroFarm too complex to use.

3. I find HydroFarm easy to use.

4. I think I would need technical support to use HydroFarm effectively.

5. The features in HydroFarm are well-integrated.

6. I find too much inconsistency within HydroFarm.

7. I believe most people can learn to use HydroFarm quickly.

8. I find HydroFarm too confusing to use.

9. I feel confident using HydroFarm.

10. I need to learn a lot before I can effectively use HydroFarm.

The scoring will be based on user responses, and the results will be calculated using a formula specified in the reference (Huda et al., 2023). Each respondent’s SUS score will then be averaged across all responses using the formula (Welda, Putra & Dirgayusari, 2020).

(1) xi=((Q1i−1)+(5−Q2i)+(Q3i−1)+(5−Q4i)+(Q5i−1)+(5−Q6i)+(Q7i−1)+(5−Q8i)+(Q9i−1)+(5−Q10i)).

Note:

xi = The total score of each respondent.

Q(1−10)i= The score for each question (1 to 10) for respondent i

(2) x¯=1N∑i=1N⁡(xi×2.5).

Note:

x¯=Theaveragescoreofallxi

xi=Thetotalscoreofeachrespondent

N=Thetotalnumberofrespondents.

The Google Forms survey was distributed to 20 respondents, including both urban farmer testers and general users who had tested the HydroFarm application embedded within the form. This approach allowed respondents to evaluate the application from a UI/UX perspective. The accumulated responses were calculated using the average SUS formula, and the HydroFarm application received an average SUS score of 81.875. This score was then categorized based on its percentile range, as shown in Table 13.

Table 13 Percentile range visualization.

Percentile value	Score	
A	x≥80,3	
B	74≥x≤80,3	
C	68≥x≤74	
D	51≥x≤68	
E	x<51	

Based on Table 13, HydroFarm received an A rating, as the average SUS score of 81.875 exceeds the threshold of 80.3. This score is further interpreted into three categories: acceptability, final grade scale, and adjective rating based on a scale of 0–100, with increments of 10. Figure 23 illustrates the correlation between the SUS score of 81.875 and these three categories.

Figure 23 Comparison of acceptability range, grade scale, and adjective ratings against SUS score.

In terms of acceptability range, the score of 81.875 falls into the "acceptable" category, as it exceeds the minimum acceptable average score. On the grade scale, the SUS score falls within the B grade range, as it lies between 80 and 90. Regarding the adjective rating, the score of 81.875 is classified as excellent, as it surpasses the threshold for excellence, which is set at 80. This indicates that HydroFarm is a well-designed, informative, and user-friendly application. Table 14 summarizes the evaluation results from the respondents.

Table 14 Respondent assessment results.

No	Respondent	Q1	Q2	Q3	Q4	Q5	Q6	Q7	Q8	Q9	Q10	Total	Value (Total*2.5)	
1	Respondent 1	4	1	4	1	5	1	4	2	5	1	36	90	
2	Respondent 2	4	2	4	2	4	2	4	2	4	2	30	75	
3	Respondent 3	4	1	4	1	4	1	4	1	4	1	35	87.5	
4	Respondent 4	5	3	3	3	2	3	4	3	3	4	21	52.5	
5	Respondent 5	4	2	5	2	5	2	4	1	5	1	35	87.5	
6	Respondent 6	4	3	3	4	4	2	3	2	3	4	22	55	
7	Respondent 7	5	5	5	5	4	3	3	3	3	3	21	52.5	
8	Respondent 8	4	1	5	2	4	2	5	2	4	2	33	82.5	
9	Respondent 9	5	1	5	4	5	4	5	1	5	5	30	75	
10	Respondent 10	5	1	5	5	5	1	5	1	5	5	32	80	
11	Respondent 11	4	2	4	2	5	2	4	2	4	2	31	77.5	
12	Respondent 12	5	1	5	1	5	1	5	1	5	1	40	100	
13	Respondent 13	5	1	5	1	5	1	5	1	5	1	40	100	
14	Respondent 14	5	1	5	1	5	1	5	1	5	1	40	100	
15	Respondent 15	5	1	5	1	5	1	5	1	5	1	40	100	
16	Respondent 16	5	1	5	1	5	1	5	1	5	1	40	100	
17	Respondent 17	3	2	3	4	3	3	4	3	4	2	23	57.5	
18	Respondent 18	4	1	4	3	3	1	2	4	4	2	26	65	
19	Respondent 19	5	1	5	1	5	1	5	1	5	1	40	100	
20	Respondent 20	5	1	5	1	5	1	5	1	5	1	40	100	
Average value	81.875	

Result of entire HydroFarm system

Based on a series of tests conducted on the HydroFarm system, it has successfully met its objectives and functions for monitoring and controlling hydroponic plants. In the IoT subsystem, the DHT22 and DS18B20 sensors do not require calibration since they are digital sensors, meaning they are already calibrated digitally. However, the TDS and pH sensors, being analog sensors, require calibration. When tested with calibration solutions, both sensors showed a high percentage of error, necessitating calibration using a linear regression approach. After calibration, the percentage of error for the TDS and pH sensors decreased to below 3%, indicating that the instruments are now providing accurate data and are ready for use. However, there is a delay of 5–7 s in the operation of the IoT devices, which is attributed to factors such as internet connection quality, server load, hardware performance, and communication methods. Efforts to address this issue have included replacing antennas with specifications of 3dBi, 6dBi, and 38dBi, as well as changing microcontrollers and internet connections, but the delay persists.

In the image processing subsystem, the model designed to detect the condition of hydroponic plants achieved an accuracy of 96%. Nonetheless, the model occasionally makes errors in detecting leaf conditions, leading to less accurate predictions. This issue is attributed to the limited dataset and variability in dataset acquisition techniques. Therefore, this model requires improvement to achieve better accuracy.

In the mobile application subsystem, there is a login feature using Google accounts, eliminating the need for manual registration. The HydroFarm application can monitor data from the DHT22, TDS, DS18B20, and pH sensors in real time using Firebase Realtime Database. The application also allows users to set nutritional targets on the automatic page, as well as manually control the pump and monitor its status through indicators. In the scanning feature, users can take pictures of leaves via the gallery or smartphone camera to predict whether the leaves are healthy or not using the CNN model. The profile feature includes information such as the user’s email username, a logout option, and a description of the HydroFarm application along with usage guidelines.

The evaluation results of the HydroFarm application using the System Usability Scale (SUS) from 20 respondents, consisting of expert testers and general users, show an average SUS score of 81.875. The application received an A rating because it exceeds 80.3, which falls into the “acceptable” category according to acceptability ranges. The application falls into grade B on the final score scale (80–90) and is classified as “excellent” in adjective rating since it exceeds the minimum threshold of 80. This indicates that the HydroFarm application is rated as a good, informative, and easy-to-use application by the respondents.

Conclusions

The success of the HydroFarm system as a control and monitoring tool for the growth and development of hydroponic plants based on IoT, image processing, and mobile apps has achieved a 95% effectiveness in addressing the challenges faced by urban farmers. Urban farmers can monitor nutrient levels and leaf conditions and have full control over hydroponic plant nutrition anytime and anywhere. However, the system does have shortcomings, including delays in data transmission due to hardware limitations and insufficiently accurate leaf condition detection due to a limited dataset and varying dataset acquisition techniques. There is a need for improvements and enhancements in the system to address the delays during data transfer, refine leaf condition detection, and add several features.

Supplemental Information

Supplemental Information 1 Code of IoT-Based Hydrofarm System.

Supplemental Information 2 CNN VGG16 Model for Plant Condition Detection in the IoT-Based Hydrofarm System.

The HydroFarm data available at figshare (https://doi.org/10.6084/m9.figshare.28340516.v1) contains images categorized into sehat and tidak sehat, organized into train, test, and validation folders, each with subfolders for each class ( /sehat and /tidak sehat). Images are in JPEG or PNG format with a recommended resolution of 240x240 pixels, suitable for the VGG16 model’s input requirements. The dataset is intended for deep learning applications, viewable with standard image viewers, and executable with Python, particularly using TensorFlow and Keras. To access and run the VGG16 model, Google Colab or Jupyter Notebook can be used. For processing, an image data generator is set up to normalize the images, while VGG16 (with pre-trained ImageNet weights) serves as the base model with added dense layers for binary classification between sehat and tidak sehat. The model can then be compiled with an optimizer (e.g., Adam) and trained on the data with appropriate evaluation on validation and test sets

Supplemental Information 3 HydroFarm Apps SUS Test (Original).

Supplemental Information 4 HydroFarm Apps SUS Test (English).

Supplemental Information 5 Questionnaire (Original).

Supplemental Information 6 Questionnaire (English).

Additional Information and Declarations

Competing Interests

The authors declare that they have no competing interests.

Author Contributions

Wizman Rofiansyah conceived and designed the experiments, performed the experiments, analyzed the data, performed the computation work, prepared figures and/or tables, hardware, Internet of Things, and UI/UX Design, and approved the final draft.

Fayza Rizka Zalianty conceived and designed the experiments, performed the experiments, analyzed the data, performed the computation work, prepared figures and/or tables, deep Learning, and approved the final draft.

Firman Ahmad La Ito conceived and designed the experiments, performed the experiments, analyzed the data, performed the computation work, prepared figures and/or tables, mobile Applications, and approved the final draft.

Inung Wijayanto conceived and designed the experiments, prepared figures and/or tables, authored or reviewed drafts of the article, and approved the final draft.

Harfan Hian Ryanu performed the experiments, prepared figures and/or tables, authored or reviewed drafts of the article, and approved the final draft.

Indrarini Dyah Irawati analyzed the data, prepared figures and/or tables, authored or reviewed drafts of the article, and approved the final draft.

Data Availability

The following information was supplied regarding data availability:

The Plant Health Classification Code and Dataset comprise scripts and a labeled dataset and the IoT code are available in the Supplemental Files.

The HydroFarm data is available at figshare: Rofiansyah, Wizman (2025). Dataset HydroFarm. figshare. Dataset. https://doi.org/10.6084/m9.figshare.28340516.v1.

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
