# Peer review of "IoT-based control and monitoring system for hydroponic plant growth using image processing and mobile applications"

_PeerJ Computer Science, doi:10.7717/peerj-cs.2763_

## Round 0.1 · original submission · Major Revisions

The manuscript requires clarification and expansion in several areas, including experimental design, results analysis, figure modifications, and addressing gaps in literature. These improvements will strengthen the overall presentation and validity of the research. The reviewers have outlined in details aspects of some of these areas of improvement. Please revise accordingly.

Reviewer 1 ·

Basic reporting

Authors detail HydroFarm, a product designed for monitoring and controlling hydroponic systems through IoT and image processing, accessible to urban farmers via a mobile application.
• Professional English used by Authors but there is scope for improvement.
• Literature references, sufficient field background/context provided. Formatting of references needed.
• Professional article structure, figures, tables. Raw data shared. Need correction at places mentioned in the report.
• Few results particularly on CNN, needs modifications.

Experimental design

Note few observations
a. The abstract does not talk about the effectiveness of the work, objective criteria to claim the success is necessary.
b. Under Materials and methods Part A section e ,The specifications of the Relay used in design are not provided.
c. Part B CNN is not explained well.Technical details are missing .This part needs revision.
Add more information related to Data set used ,No of images,size of image ,no of epochs,Batch size,Activation function.
Modify the sentences on line no 205,206
d. User interface is quite good for Mobile Application
e. Figure Wiring of IoT Devices for HydroFarm System needs to be modified
Add annotations
f. Figure 4 Block Diagram of CNN Training Process is not satisfactory
• The preprocessing step could be broken down further to specify what preprocessing entails.
• The "training dataset" block can be renamed. For example, you could specify "Model Training" to reflect the actual activity.
• Add clarity to "Is the model suitable?" by indicating what suitability means ?
• The "errors in the code, number of layers, or training process" block could be more detailed, or the wording could be refined to indicate debugging and hyperparameter tuning.
g. Figure5 Block Diagram of CNN Implementation on Mobile Apps
• It is not at all as per the title, need to be updated.
h. Figure 13 Confusion Matrix from First Test
Need to correct for Labels
i. Figure 14,15,16 Results


j. Under Results of The Internet of Things (IoT) System section
• The steps given on 290-292 and similarly for other sensors are unnecessarily increasing the length of paper.

Validity of the findings

2. VALIDITY OF THE FINDINGS
• Research question well defined, relevant & meaningful.
• Knowledge gap is not identified. Literature related to the topic is available and should be referred to
improve the contents.
• Rigorous investigation performed for Sensor testing but it is missing for CNN part.

Additional comments

The length of the paper can be reduced .

·

Basic reporting

a. In the abstract, I suggest that the author highlight their paper's potential so that the novelty of the paper can easily target the research audience.
b. No literature research on the given topic is highlighted in the paper. So add some literature review in the manuscript.
c. Figures should have the proper labelling (x and y axis).

Experimental design

a. As mentioned in the section (Materials & Methods) you have used the ESP32 for connecting all the sensors and pump (as shown in Figure 2). So was the ESP32 able to drive all the sensors simultaneously?
b. Also, in Figure 2 you have used the LCD screen (which displays www.sunfounder.com) You can use your figures.
c. Figure 3 PCB Design for Internet of Things. What this PCB is denoting is not understood. Need details explanation.
d. You mention the Raspberry Pi camera for capturing the photo. However, I cannot find the Raspberry Pi camera in your hardware setup.
e. You also mentioned that your application model uses the CNN model to detect the health condition of your plant. (So, how many days have you recorded the plant growth).
f. On what parameters do you provide the nutrients to your plant? And how your setup decides it (managing the nutrients in the hydroponic plants).
g. As shown in Figure 7. Two plants are shown in the setup. So the nutrition required for these plants are same or different.
h. The author used the VGG16 network for the health detection of the leaf. Now the question is why to use the VGG16 network for such small datasets. It will only increase the complexity of the system. It needs to be justified.

Validity of the findings

a. The novelty of the paper is missing.
b. The results presented in the manuscript do not focus on the discussion rather than only on the reading obtained using the sensors ( example Figures 9/10)
c. No comparison is been made with the natural growth of the plant with the proposed work which is mainly lacking in the manuscript.
d. The SUS score has been calculated. Using the 20 respondents from the Google sheet. I think it is more useful when you deploy your Mobile App with UX point of view. It is quite natural that you will be getting a high score since the survey you are conducting within your lab setups.
e. From Table 1-12 the results obtained from the S.No 1-10 are same and then you are taking the average. So, it is not clear. Why the data is not changing for any S.N. It has to be addressed and you can reduce the numbers of tables.

---

## Round 0.2 · accepted · Accept

I have reviewed the revision (and the latest reviews) and am satisfied that you have adequately addressed the concerns raised by the reviewers to the extent that is feasible.

Reviewer 1 ·

Basic reporting

'no comment

Experimental design

The data set used has very few imges which questions the validity of results.
It is a project implementaion and not a indepth research.
Significant novelty is not visible.

Validity of the findings

The impact and novelty accessment is not as expected by a research article.

Additional comments

I did not receive the answers related to comments given in prior review.
The revision done is not as expected.

·

Basic reporting

The author has revised the manuscript according to the comment. Also, all the comments are addressed.

Experimental design

no comment

Validity of the findings

no comment

Additional comments

Few Suggestion.
1. Figure 12, Figure 14, Figure 15 and Figure 17 can be shown with the table (No need to place the screenshot).